# Analysis of the Association between Female Medical History and Thyroid Cancer in Women: A Cross-Sectional Study Using KoGES HEXA Data

**DOI:** 10.3390/ijerph18158046

**Published:** 2021-07-29

**Authors:** Young Ju Jin, Suk Woo Lee, Chang Myeon Song, Bumjung Park, Hyo Geun Choi

**Affiliations:** 1Department of Otorhinolaryngology-Head & Neck Surgery, Wonkwang University Hospital, Wonkwang University College of Medicine, Iksan 54538, Korea; chindol1@wku.ac.kr; 2Department of Obstetrics and Gynecology, Hallym University College of Medicine, Anyang 14068, Korea; ssugucap@hallym.ac.kr; 3Department of Otolaryngology-Head and Neck Surgery, Hanyang University College of Medicine, Seoul 04763, Korea; cmsong@hanyang.ac.kr; 4Department of Otorhinolaryngology-Head & Neck Surgery, Hallym University College of Medicine, Anyang 14068, Korea; pbj426@hallym.ac.kr; 5Hallym Data Science Laboratory, Hallym University College of Medicine, Anyang 14068, Korea

**Keywords:** thyroid cancer, hysterectomy, oophorectomy, oral contraceptives, number of children, cross-sectional study

## Abstract

The purpose of this study was to evaluate the association between female medical history and thyroid cancer. **Methods:** Data from the Korean Genome and Epidemiology Study were collected from 2004 to 2016. Among a total of 1303 participants with thyroid cancer and 106,602 control (non-thyroid cancer) participants, the odds ratios (ORs) with 95% confidence intervals (CIs) of hysterectomy, oophorectomy, use of oral contraceptives, and number of children were evaluated. **Results:** The adjusted OR of hysterectomy for thyroid cancer was 1.73 (95% CI = 1.48–2.01, *p* < 0.001) in the minimally adjusted model. The adjusted ORs for thyroid cancer were 1.89 (95% CI = 1.06–3.37, *p* = 0.031), 0.89 (95% CI = 0.83–0.94, *p* < 0.001), and 0.85 (95% CI = 0.73–0.99, *p* = 0.040) for bilateral oophorectomy, number of children, and use of oral contraceptives, respectively, in the fully adjusted model. In the subgroup analysis, the adjusted ORs of bilateral oophorectomy were significant in the younger age (OR = 3.62, 95% CI = 1.45–9.03, *p* = 0.006), while the number of children was significant in the older age (OR = 0.86, 95% CI = 0.80–0.93, *p* < 0.001). **Conclusions:** The ORs of hysterectomy and bilateral oophorectomy were significantly higher in the thyroid cancer group in the younger age group. The adjusted ORs of the number of children were significantly low in the older age group.

## 1. Introduction

Thyroid cancer is the second most common female cancer in Korea after breast cancer. In the United States, the incidence of thyroid cancer increased from 4.56 cases per 100,000 to 14.42 per 100,000 individuals from 1975 to 2013 [1], and, in Korea, the incidence of thyroid cancer increased sharply from 7.2 cases per 100,000 to 51.0 per 100,000 individuals from 1994 to 2016. Although progress has been made in the diagnosis and treatment of thyroid cancer, the etiology of thyroid cancer is still unclear. Several studies have suggested that the development of thyroid cancer is related to genetic factors, environmental pollutants, lifestyle, obesity, sex hormones, radiation exposure, elevated TSH levels, history of proliferative thyroid disease, iodine enrichment, and diet [2,3,4,5].

Generally, the incidence of thyroid cancer is 3–5 times higher among women than men. The incidence of thyroid cancer among women increases during their reproductive years from 4% at 10–19 years of age to 30–39% at 30–39 years of age, and declines after menopause to 17.4% at ≥50 years of age. However, it increases steadily in men with age [6,7]. Therefore, female sex hormones have been suggested to play a role in the increase in thyroid cancer, as estrogens have been reported to have direct actions on proliferative and neoplastic disorders [8]. Three estrogens (E1, E2, E3) and two estrogen receptors (ERα, ERβ) play important roles in the proliferation and invasion of thyroid cancer [8,9]. However, epidemiologic studies have shown that there is controversy regarding the association between pregnancy, parity, menstrual cycle regularity, oral contraceptive (OC) use, hysterectomy, oophorectomy, menopausal status, and thyroid cancer risk [4,10,11]. Thus, we hypothesized that there is a link between a woman’s reproductive history and the risk of thyroid cancer.

The aim of this study was to evaluate the association between female medical history and thyroid cancer based on the Korean Genome and Epidemiology Study (KoGES) health examinee (HEXA) data. The odds ratios (ORs) of hysterectomy, oophorectomy, number of children, and use of oral OCs for thyroid cancer compared to a control group were evaluated, with additional subgroup analyses.

## 2. Materials and Methods

### 2.1. Study Population and Data Collection

The ethics committee of Hallym University (20 February 2019) approved the use of these data. The requirement for written informed consent was waived by the Institutional Review Board. This prospective cohort study relied on data from the KoGES (4851-302), National Research Institute of Health, Centers for Disease Control and Prevention, Ministry for Health and Welfare, Republic of Korea from 2004 to 2016. A detailed description of these data is provided in a previous study [12]. From the KoGES Consortium, we used KoGES health examinee (HEXA) data consisting of participants ≥ 40 years old who resided in urban areas. The dataset consisted of the base data from 2004 to 2013 and follow-up data from 2012 to 2016.

### 2.2. Participant Selection

Among 173,209 participants, we excluded men (*n* = 59,261), participants who lacked records of cancer histories (*n* = 594), participants who lacked records on the use of OCs (*n* = 1981), participants who smoked (*n* = 557), participants with body mass index (BMI) data (*n* = 383), and participants with histories of hypertension, diabetes mellitus, and thyroid disease (*n* = 62). We excluded participants with other cancer histories (gastric cancer, hepatic cancer, colon cancer, breast cancer, lung cancer, cervical cancer, or bladder cancer, *n* = 3006). Then, 1303 thyroid cancer participants and 106,602 control (no thyroid cancer) participants were selected (Figure 1).

### 2.3. Survey

Each participant was asked about her previous history of cancer, including thyroid cancer, gastric cancer, hepatic cancer, colon cancer, breast cancer, lung cancer, cervical cancer, or bladder cancer, by trained interviewers. Each was also asked her medical history of hypertension, diabetes mellitus, any thyroid disease (hyperthyroidism or hypothyroidism), hysterectomy, oophorectomy (unilateral or bilateral), use of OCs, and number of children. BMI was calculated by kg/m^2^ using the health checkup data. Regarding smoking history, women were categorized as nonsmokers (<100 cigarettes in her entire life), past smokers (quit more than one year before), or current smokers, and, regarding alcohol consumption, women were categorized as nondrinkers, past drinkers, or current drinkers. Occupation was classified into 10 standard Korean occupations: manager; expert; specialist; clerk; service worker; salesperson; farmer or fisherman; technician, mechanic, production worker, or engineer; laborer; and soldier. Then, women were recategorized into 4 groups in this study: (1) manager, expert, specialist, or clerk; (2) service worker or salesperson; (3) farmer, fisher, technician, mechanics, production worker, or engineer; and (4) homemaker or unemployed.

### 2.4. Statistical Analyses

Chi-square tests were used to compare the rates of hypertension, diabetes mellitus, any thyroid disease, occupation, smoking status, alcohol consumption, hysterectomy, oophorectomy, and use of OCs. Independent T-tests were used for comparisons of age, BMI, and number of children.

To analyze the ORs of hysterectomy, oophorectomy, use of OCs, and number of children for thyroid cancer, a logistic regression model was used. In the crude model, we inserted only hysterectomy, oophorectomy, use of OCs, or number of children as the independent variable. In model 1, we added age, BMI, hypertension, diabetes mellitus, history of any thyroid disease, occupation, smoking status, or alcohol consumption as the independent variable. In model 2, we adjusted for model 1 plus hysterectomy, oophorectomy, use of OCs, and number of children. Because the history of hysterectomy and the history of oophorectomy were closely related (Appendix A), we removed the hysterectomy variable from model 2 in the final model. The 95% confidence interval (CI) was calculated. For the subgroup analysis, we divided the participants by age according to the median value.

Two-tailed analyses were conducted, and P values less than 0.05 were considered to indicate significance. The results were statistically analyzed using SPSS v. 24.0 (IBM, Armonk, NY, USA).

## 3. Results

### 3.1. Detailed Descriptions

#### 3.1.1. General Characteristics of Participants

The proportions of thyroid disease (19.9% vs. 6.2%), history of hysterectomy (17.0% vs. 10.4%), and history of bilateral oophorectomy (0.9% vs. 0.5%) were significantly higher in the thyroid cancer group than in the control group (Table 1). On the other hand, the proportions of current smokers (0.8% vs. 1.3%), past smokers (1.4% vs. 2.4%), current drinkers (26.7% vs. 31.5%), and users of OCs (15.3% vs. 18.1%) were lower among thyroid cancer patients than among controls. The mean number of children was higher in the control group (*n* = 2.28) than in the thyroid cancer group (*n* = 2.15) (Table 1).

#### 3.1.2. ORs of Hysterectomy, Oophorectomy, the Number of Children, and Use of Oral Contraceptive for Thyroid Cancer

The adjusted ORs of hysterectomy and bilateral oophorectomy for thyroid cancer were significantly increased to 1.73 (95% CI = 1.48–2.01, *p* < 0.001) in model 2 and 1.89 (95% CI = 1.06–3.37, *p* = 0.031) in the final model. The adjusted ORs of the number of children and use of OCs for thyroid cancer were significantly decreased to 0.89 (95% CI = 0.83–0.94, *p* < 0.001) and 0.85 (95% CI = 0.73–0.99, *p* = 0.040), respectively, in the final model (Table 2).

#### 3.1.3. ORs of Hysterectomy, Oophorectomy, the Number of Children, and Use of Oral Contraceptive for Thyroid Cancer According to Age

The subgroup analysis was classified into younger and older age groups compared to the control group. In the younger age group, the adjusted ORs of hysterectomy and bilateral oophorectomy for thyroid cancer were significantly increased to 1.54 (95% CI = 1.19–1.99, *p* = 0.001) in model 2 and 3.62 (95% CI = 1.45–9.03, *p* = 0.006) in the final model. In the older age group, the adjusted OR of hysterectomy for thyroid cancer compared to the control group was 1.74 (95% CI = 1.44–2.10, *p* < 0.001) in model 2. The adjusted OR of the number of children for thyroid cancer was significantly decreased to 0.86 (95% CI = 0.80-0.93, *p* < 0.001) in the final model (Table 3).

## 4. Discussion

In this study, the adjusted ORs of hysterectomy and bilateral oophorectomy were significantly higher in the thyroid cancer group than in the control group. The adjusted ORs of the number of children and use of OCs were significantly reduced in the thyroid cancer group compared with the control group. In the subgroup analysis, the adjusted OR of bilateral oophorectomy was significantly higher in the younger age thyroid cancer group than in the younger age control group. The adjusted ORs of the number of children and use of OCs were significantly decreased in the older age thyroid cancer group compared to the older age control group.

In this study, the ORs of bilateral oophorectomy were significantly higher in the thyroid cancer group than in the control group. Surgical menopause was suggested to have a positive association with thyroid cancer compared to natural menopause. A European prospective study analyzed 508 thyroid cancer cases and found that thyroid cancer risk was significantly increased in participants with artificial menopause compared to natural menopause (HR: 2.16, 95% CI = 1.41–3.31) [10]. In a Japanese prospective study, the risk of thyroid cancer was significantly increased in cases with surgical menopause compared to natural menopause (HR: 2.34, 95% CI = 1.43–3.84) [13]. In a meta-analysis, three of five case–control studies showed a very high significant increase in the risk of thyroid cancer in an artificial menopause group compared with the natural menopause group (OR: 2.05, 95% CI = 1.39–3.01) [14]. The mechanisms of association between surgical menopause and the elevated risk of thyroid cancer were not fully investigated. Possible causes are as follows. First, the sudden drop in estrogen levels in oophorectomy can significantly affect the functioning of the thyroid gland, which could increase the risk of thyroid cancer compared with natural menopause [8,14]. Second, a history of transfusion during oophorectomy could increase the risk of thyroid cancer compared to the general population (RR: 1.77, 95% CI = 0.95–3.30) [15]. Immunosuppression by transfusion promotes the carcinogenic progression from thyroiditis to cancer, and could increase the rate of cancer recurrence [16]. Third, after surgical menopause, women could be more carefully monitored for any hormonal change, including thyroid imbalances, based on regular follow-up. Therefore, there is a possibility of surveillance bias [17].

The ORs of the use of OCs were decreased in the thyroid cancer group compared with the control group, especially for the older age group. However, evidence for a link between OCs and thyroid cancer risks is inconsistent. Many studies support the notion that the risk of papillary thyroid cancer is reduced in women who had ever used OCs [10,13,18,19,20]. The prospective US study showed that OC use for ≥10 years (vs. never used) was inversely associated with thyroid cancer risk (HR: 0.48, 95% CI = 0.28–0.84, P = 0.01) [19]. In a prospective European cohort study, OC use showed a significant inverse association with the risk of differentiated thyroid cancer (HR: 0.48, 95% CI = 0.25–0.92) [10]. A meta-analysis of prospective cohort studies found a significant inverse association between OC use and the risk of thyroid cancer by analyzing nine studies with a total of 1,906 patients (RR: 0.84, 95% CI = 0.73–0.97) [20]. However, there are many studies without a significant association between OC use and thyroid cancer risk [21,22,23,24]

The possible causes of inconclusive results regarding the risk of thyroid cancer and OC use are as follows. First, the participant’s age could lead to different results. In our study, although the overall ORs of OC for thyroid cancer risk were significantly decreased, there was no significant difference after classification according to age. Because the incidence of thyroid cancer peaks among reproductive-aged women, evaluation for the reproductive age group is essential to determine the real association between OC use and thyroid cancer risk [25]. Similar to our study results, previous studies did not find a link between OC use and thyroid cancer risk when evaluating reproductive-aged women [24,26]. Second, differences in the formulations of OCs could lead to inconclusive study results. The formulation has been changed to lowered ethinyl estradiol content, with the introduction of 17ß estradiol, and inclusion of various kinds of progestin, although the standard regimen of steroid-containing pills continues to be applied for 21 days, with a pill-free interval of 7 days [27]. Third, OC use and duration were investigated by self-questionnaire, and ‘recall bias’ could affect much older women more than young women.

The ORs of the number of children were decreased in the thyroid cancer group compared with those in the control group, especially for the older age group. In support of our results, two previous studies performed in Japan and Thailand found that thyroid cancer risk was decreased among women who had experienced pregnancy compared with those who had not [28,29]. However, many previous studies suggested that the number of pregnancies and parity could increase the risk for thyroid cancer [6,12,30,31,32]. Additionally, some studies have suggested no significant association between the number of pregnancies and the risk of thyroid cancer [4,10,11].

In our study, the risk of thyroid cancer was decreased when the number of children increased. The proliferation of thyroid tumors is influenced by the female sex hormone, estrogen. There are three kinds of estrogen: E1 (estrone), produced in adipose tissue and the adrenal glands; E2 (17β-estradiol), produced in the ovaries; and E3, produced by the placenta during pregnancy. Estrone and E3 are metabolites of E2, whose levels are highest among the estrogens, and which has the highest affinity to estrogen receptors. E2 binds directly with ERα and ERβ in thyroid cancer cells. The proliferation of thyroid cancer cells is promoted by ERα, whereas apoptosis is controlled by the enhanced expression of ERβ [9]. An increased number of pregnancies could decrease E2 exposure by reducing lifetime menstrual cycling. Therefore, shorter exposure to E2 could decrease the risk of thyroid cancer in multiparous women [33,34]. It is reasonable to explain why the ORs of the number of children for thyroid cancer were significantly decreased not in the younger age group, but in the older age group.

There are several limitations. First, there was a lack of information on several potential confounding factors, such as ionizing radiation exposure, specific formulation of OCs, family history of thyroid cancer, and iodine deficiency, which are well-known thyroid cancer risk factors. Second, we did not have information on the patients’ menopausal status at diagnosis of thyroid cancer. Third, information about menopausal status at diagnosis of thyroid cancer and the pathologic type were not included in the analysis. Fourth, different with previous studies, the proportion of thyroid disease was significantly higher in the thyroid cancer group than in the control group. This result could be caused by increased surveillance or diagnostic evaluation in the thyroid cancer group.

## 5. Conclusions

Our study provides very important evidence of the relationship between reproductive factors and thyroid cancer risk. The adjusted ORs of oophorectomy were significantly higher in the thyroid cancer group than in the control group, especially in the younger age group. The adjusted ORs of the number of children and use of OCs were significantly reduced in the thyroid cancer group compared with the control group, especially in the older age group. Further study is needed on circulating thyroid-stimulating hormone and sex hormones, such as estrogen and progesterone.

## Figures and Tables

**Figure 1 ijerph-18-08046-f001:**
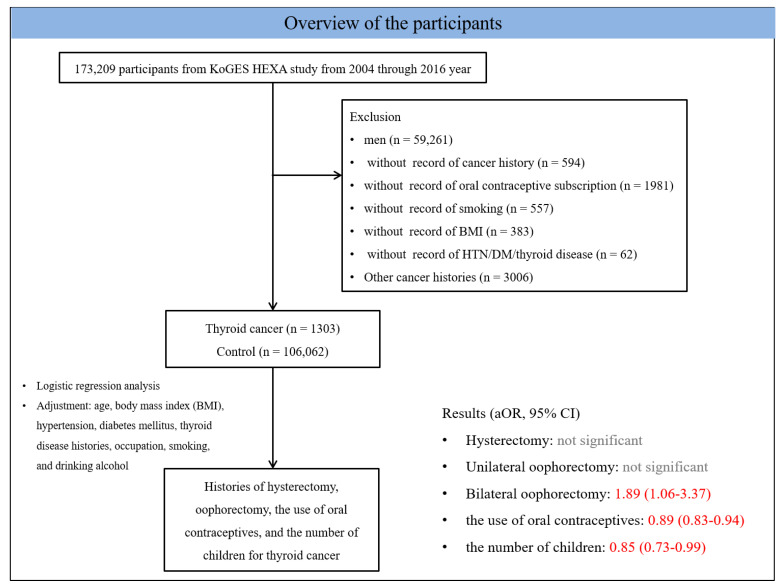
A schematic illustration of the participant selection process that was used in the present study. Of a total of 173,209 participants, 1303 in the thyroid cancer group and 106,062 in the control group were evaluated. For them, the risk of thyroid cancer was estimated for possible risk factors, including history of hysterectomy, oophorectomy, use of OCs and number of children after adjustments for age, BMI, hypertension, diabetes mellitus, thyroid disease history, occupation, smoking status, and drinking consumption.

**Table 1 ijerph-18-08046-t001:** General characteristics of participants.

Characteristics		Thyroid Cancer Histories		*p*-Value
		Yes	No	
Number (*n*, %)		1303 (1.2)	106,062 (98.8)	
Age (y, mean, SD)		52.2 (7.1)	52.6 (8.1)	0.040*
BMI (kg/m^2^, mean, SD)		23.8 (2.9)	23.7 (3.0)	0.210
Hypertension (*n*, %)		242 (18.6)	18,825 (17.7)	0.439
Diabetes mellitus (*n*, %)		60 (4.6)	5592 (5.3)	0.283
Thyroid disease (*n*, %)		259 (19.9)	6546 (6.2)	<0.001 *
Occupation (*n*, %)				0.001 *
	Manager, expert, specialist, clerk	243 (18.6)	18,072 (17.0)	
	Service worker, sales person	185 (14.2)	17,273 (16.3)	
	Farmer, fisher, technician, mechanics, production worker	111 (8.5)	12,040 (11.4)	
	Housemaker, unemployed	764 (58.6)	58,677 (55.3)	
Smoking (*n*, %)				0.021 *
	Non-smoker	1274 (97.8)	102,169 (96.3)	
	Past smoker	18 (1.4)	2520 (2.4)	
	Current smoker	11 (0.8)	1373 (1.3)	
Drinking alcohol (*n*, %)				0.001 *
	Non-drinker	926 (71.1)	70,516 (66.5)	
	Past drinker	29 (2.2)	2,175 (2.1)	
	Current drinker	348 (26.7)	33,371 (31.5)	
Hysterectomy (*n*, %)		221 (17.0)	11,074 (10.4)	<0.001 *
Oophorectomy (*n*, %)				0.101
	No	1274 (97.8)	104,135 (98.2)	
	Unilateral oophorectomy	17 (1.3)	1398 (1.3)	
	Bilateral oophorectomy	12 (0.9)	529 (0.5)	
Use of oral contraceptive (n, %)		199 (15.3)	19183 (18.1)	0.009 *
The number of children (mean, SD)		2.15 (0.89)	2.28 (1.06)	<0.001 *

* Independent T-test or chi-square test, statistical significance at *p* < 0.05.

**Table 2 ijerph-18-08046-t002:** Odds ratios of hysterectomy, oophorectomy, the number of children, and use of oral contraceptive for thyroid cancer.

Variables		ORs for Thyroid Cancer							
		Crude	*p*-Value	Model 1 ^†^	*p*-Value	Model 2 ^‡^	*p*-Value	Final ^§^	*p*-Value
Hysterectomy		1.75 (1.51–2.03)	<0.001 *	1.75 (1.51–2.04)	<0.001*	1.73 (1.48–2.01)	<0.001*		
Oophorectomy									
	Unilateral	0.99 (0.61–1.61)	0.980	0.98 (0.61–1.59)	0.936	0.76 (0.46–1.23)	0.261	0.96 (0.59–1.55)	0.858
	Bilateral	1.85 (1.04–3.30)	0.035 *	1.93 (1.09–3.45)	0.025 *	1.28 (0.71–2.31)	0.412	1.89 (1.06–3.37)	0.031 *
The number of children		0.89 (0.84–0.94)	<0.001 *	0.88 (0.83–0.94)	<0.001 *	0.90 (0.84–0.95)	<0.001 *	0.89 (0.83–0.94)	<0.001 *
The use of oral contraceptive		0.82 (0.70–0.95)	0.009 *	0.84 (0.72–0.98)	0.026 *	0.85 (0.73–0.99)	0.041 *	0.85 (0.73–0.99)	0.040 *

* Logistic regression was analyzed, significance at *p* < 0.05. † Model 1 was adjusted for age, body mass index (BMI), hypertension, diabetes mellitus, thyroid disease histories, occupation, smoking, and drinking alcohol. ‡ Model 2 was adjusted for Model 1 plus hysterectomy, oophorectomy, the number of children, and the use of oral contraceptive. § Final model was adjusted for Model 1 plus oophorectomy, the number of children, and the use of oral contraceptive.

**Table 3 ijerph-18-08046-t003:** Subgroup analyses of odds ratios of hysterectomy, oophorectomy, the number of children, and use of oral contraceptive for thyroid cancer according to age.

Variables		ORs for Thyroid Cancer							
		Crude	*p*-Value	Model 1 ^†^	*p*-Value	Model 2 ^‡^	*p*-Value	Final ^§^	*p*-Value
**Younger Age Group (*n* = 51,366)**									
Hysterectomy		1.74 (1.36–2.22)	<0.001*	1.60 (1.24–2.05)	<0.001*	1.54 (1.19–1.99)	0.001*		<0.001*
Oophorectomy									
	Unilateral	1.34 (0.69–2.59)	0.393	1.24 (0.64–2.43)	0.521	1.09 (0.55–2.13)	0.814	1.25 (0.64–2.44)	0.513
	Bilateral	3.80 (1.55–9.36)	0.004 *	3.61 (1.45–8.99)	0.006 *	2.72 (1.07–6.90)	0.036 *	3.62 (1.45–9.03)	0.006 *
The number of children		1.06 (0.96–1.18)	0.265	1.03 (0.92–1.15)	0.597	1.04 (0.93–1.16)	0.524	1.03 (0.92–1.15)	0.573
The use of oral contraceptive		0.88 (0.69–1.12)	0.287	0.89 (0.70–1.14)	0.359	0.89 (0.70–1.13)	0.346	0.89 (0.70–1.13)	0.352
**Older Age Group ( *n* = 55,999)**									
Hysterectomy		1.77 (1.48–2.13)	<0.001 *	1.72 (1.43–2.07)	<0.001 *	1.74 (1.44–2.10)	<0.001 *		<0.001 *
Oophorectomy									
	Unilateral	0.77 (0.38–1.55)	0.463	0.74 (0.37–1.49)	0.401	0.55 (0.27–1.11)	0.095	0.72 (0.36–1.45)	0.359
	Bilateral	1.35 (0.64–2.85)	0.439	1.43 (0.67–3.03)	0.356	0.93 (0.43–1.99)	0.842	1.38 (0.65–2.94)	0.400
The number of children		0.80 (0.75–0.86)	<0.001 *	0.86 (0.79–0.93)	<0.001 *	0.87 (0.80–0.94)	<0.001 *	0.86 (0.80–0.93)	<0.001 *
The use of oral contraceptive		0.77 (0.63–0.94)	0.011 *	0.82 (0.67–1.00)	0.045 *	0.83 (0.68–1.02)	0.076	0.83 (0.68–1.02)	0.075

* Logistic regression was analyzed, significance at *p* < 0.05. † Model 1 was adjusted for age, body mass index (BMI), hypertension, diabetes mellitus, thyroid disease histories, occupation, smoking, and drinking alcohol. ‡ Model 2 was adjusted for Model 1 plus hysterectomy, oophorectomy, the number of children, and the use of oral contraceptive. § Final model was adjusted for Model 1 plus oophorectomy, the number of children, and the use of oral contraceptive.

## Data Availability

Release of the data by the authors is not legally allowed. Data in this study are available on the database of KoGES HEXA database in NIH https://nih.go.kr/ (assessed on 11 February 2021). NIH permits access to all of these data via download for any researcher who promises to follow the research ethics.

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
