# Peer review of "Analysis of the Association between Female Medical History and Thyroid Cancer in Women: A Cross-Sectional Study Using KoGES HEXA Data"

_ijerph, 2021, doi:10.3390/ijerph18158046_

Round 1

Reviewer 1 Report

Dear Author,

in section results, table 1 - there is no information about p value for each factor, why?

the disscussion section is schematic. The same sentences repeat in each paragraph. 
I’m not sure if the author should include such strong information about strengths of the arcticle. In my opinion this should be in the cover letter not in the disscussion.

finnaly, the weak points of article are important in the context of results. I am thinking about menopausal status at diagnosis of thyroid Cancer and the pathologic type of Cancer. 

Reviewer 2 Report

I read with interest Young Ju Jin article concerning the link between a woman’s reproductive history and the risk of thyroid cancer.

Thyroid cancer has been related to various factors beside them sex hormones have been suggested to play a role in the increase the incidence of this cancer.

The authors evaluate the association between female medical history and thyroid cancer based on data of the KoGES and the HEXA and they found that hysterectomy, oophorectomy, number of children and use of oral OCs could be a risk factor for developing thyroid cancer compared to a control group.

This article is well written and interesting. Anyway, as mentioned in limitations by the author there are several potential confounding factors that have not been evaluated this would have increased the scientific soundness and the impact of this article.

Reviewer 3 Report

This is a very interesting study about the influence of hormonal changes in female patients with thyroid cancer compared to a control group. The numbers are impressive. 

It is an important topic that should be taken into account when treating with thyroid cancer Patients.  The only suggestion for adaptation of the manuscript I have is the term „thyroid disease“ that should be elucidated because it does not lead to a higher risk of cancer development but a higher chance of diagnosing it.  furthermore I would suggest a comment about regional differences in cancer prevealence that possibly does not allow to imply these findings on other countries. 

Round 2

Reviewer 1 Report

Dear Author,

thank you for correction of manuscript, it can be published in prezent form.